# Multilingual Pre-training
# with Universal Dependency Learning

**Kailai Sun**[†], **Zuchao Li**[†], **Hai Zhao**[*]
[1]Department of Computer Science and Engineering, Shanghai Jiao Tong University
[2]Key Laboratory of Shanghai Education Commission for Intelligent Interaction
and Cognitive Engineering, Shanghai Jiao Tong University, Shanghai, China
[3]MoE Key Lab of Artificial Intelligence, AI Institute, Shanghai Jiao Tong University
{kaishu2.0,charlee}@sjtu.edu.cn, zhaohai@cs.sjtu.edu.cn

## Abstract

The pre-trained language model (PrLM) demonstrates domination in downstream natural language processing tasks, in which multilingual PrLM takes advantage of language universality to alleviate the issue of limited resources for low-resource languages. Despite its successes, the performance of multilingual PrLM is still unsatisfactory, when multilingual PrLMs only focus on plain text and ignore obvious universal linguistic structure clues. Existing PrLMs have shown that monolingual linguistic structure knowledge may bring about better performance. Thus we propose a novel multilingual PrLM that supports both explicit universal dependency parsing and implicit language modeling. Syntax in terms of universal dependency parse serves as not only pre-training objective but also learned representation in our model, which brings unprecedented PrLM interpretability and convenience in downstream task use. Our model outperforms two popular multilingual PrLM, multilingual-BERT and XLM-R, on cross-lingual natural language understanding (NLU) benchmarks and linguistic structure parsing datasets, demonstrating the effectiveness and stronger cross-lingual modeling capabilities of our approach.

## 1 Introduction

The pre-trained language model (PrLM) such as BERT [1] and many kinds of its variants [2, 3, 4] have proved their effectiveness in many downstream natural language processing (NLP) tasks including semantic textual similarity [5], question answering [6], sentiment classification [7], linguistic structure [4, 8] and so on. Most of these PrLM are aimed at languages that with a large amount of available linguistic resources and are widely used, such as English. However, it is not realistic to train an individual PrLM for all languages, especially for those low-resource languages. As a result, several multilingual PrLMs which take advantage of language universality have been published and shown good cross-lingual performance on several NLP tasks.

Despite its successes, the unsupervised method typically used by multilingual PrLMs makes cross-lingual transfer inefficiency and keeps the learning still challenging. Improvement can be made by adding explicit cross-lingual signals including bitext (XLM) [9] and word translation pairs from a dictionary [10]. This suggests that the effectiveness of multilingual PrLM can be further improved by integrating explicit universal linguistic characteristics. Existing PrLMs [11, 12] have tried to incorporate monolingual linguistic structure knowledge to improve the performance across multiple linguistics tasks by Multi-Task Learning (MTL) [13]. However, the combination of universal

---

[*]Corresponding author. [†] These authors made equal contribution. This work was supported by Key Projects of National Natural Science Foundation of China (U1836222 and 61733011), Project from Chinese National Key Laboratory of Science and Technology on Information System Security.

35th Conference on Neural Information Processing Systems (NeurIPS 2021).

linguistic structure knowledge has not been explored in the multilingual area. Learning universal knowledge across languages is more complex than learning monolingual knowledge, so a better integrating method than MTL needs to be explored.

Syntactic dependency parsing disclosing syntactic relations between words in a sentence, has been found to be extremely useful for many NLP tasks [14, 15, 16]. The syntactic dependency parsing is also limited by low-resource languages. To meet the huge demand for training syntactic parser among various languages, the project of universal dependencies (UD) Treebanks was launched [17] which provides a uniform syntactic parsing structure for different languages. Therefore, UD offers an excellent universal structure characteristic, which is worth exploiting for the multilingual PrLM.

In this paper, we propose a multilingual PrLM that supports both explicit universal dependency parsing and implicit language modeling. Unlike using MTL in monolingual works, we embed a parsing scorer in our PrLM, and directly optimizes this scorer and the encoders below it with UD pre-training; meantime, we propose a structural encoder to encode the predicted structure given by the parsing scorer and integrated it into the final representation for other pre-training or downstream training process. Our approach can be smoothly applied to a variety of multilingual PrLM such as Multilingual-BERT (m-BERT) [1] and XLM-R [18].

To verify the cross-lingual modeling capabilities of our model, we carry on the experiments on both cross-lingual NLU benchmarks: XNLI and XQuAD, and linguistic structure parsing datasets: UD[2] v2.7, SPMRL'14 [19], English Penn Treebank (PTB) 3.0 [20] and the Chinese Penn Treebank (CTB) 5.1 [21]. Our empirical results show that universal structure knowledge learnt and integrated can indeed help the multilingual PrLM obtain better universal linguistic word representations and outperform m-BERT and XLM-R baselines in all the above tasks.

## 2 Related Work

**Monolingual PrLM with structure learning**    Previous works have tried to improve monolingual PrLM by learning linguistic structure [2, 12, 22, 23, 24, 25]. Some of them use plain text to implicitly learn structural knowledge, such as StrucBERT [22], which lets the PrLM encode the dependency between sentences and words by adding the word-level ordering and sentence-level ordering objectives during pre-training. However, this may not be suitable for multilingual setting. Since there are obvious differences in the word order of different languages, putting disordered plain text of multiple languages together for common training will lead to confusion in implicitly learning structural information. Contrastively, we use annotated syntactic structures knowledge in UD that provide clear guidance for improving cross-lingual representation in multilingual PrLM.

Other works use annotated syntactic knowledge as the structural knowledge such as LIMIT-BERT [12] and LISA [24]. LIMIT-BERT is a monolingual PrLM that achieves advancement in several parsing datasets and NLU tasks by performing MTL on multiple linguistic tasks including syntactic parsing. The main difference between LIMIT-BERT and our method is that we do not regard parsing as a pre-training objective, but use the parsing component as an intermediate structure to explicitly learn and extract of the syntactic structure, thereby reducing the black box characteristics of the model. LISA is a Transformer model that uses syntax information to enhance SRL. It incorporates syntactic information by training one attention head predicting syntactic dependency arc. First of all, LISA only aims at one specific task, so it is doubtful whether its method can be applied to the PrLM and improve the performance of multiple tasks. Secondly, LISA can only use the arc feature of syntax but not the relations, while our approach of syntactic structural integration can integrate both of the arc and relation features learned by our model into the final universal representation. By the way, our PrLM can be used directly as a well-behaved parser, but LISA clearly does not.

**Multilingual PrLM with explicit cross-lingual signals**    Cross-lingual pre-training tasks including cross-lingual word recovery, cross-lingual paraphrase classification and cross-lingual masked language model can take advantage of bitext to learn the mappings among different languages from more perspectives [26]. Recently, Ahmad et al. [27] proposes a multilingual PrLM training mBERT using an auxiliary objective to encode the universal dependency tree structure that helps cross-lingual transfer. However, our work significantly differs from theirs. We regard parsing not only as a pre-training objective, but also as an active part of the overall PrLM structure, so this part

---
[2]`https://lindat.cz/repository/xmlui/handle/11234/1-3424`

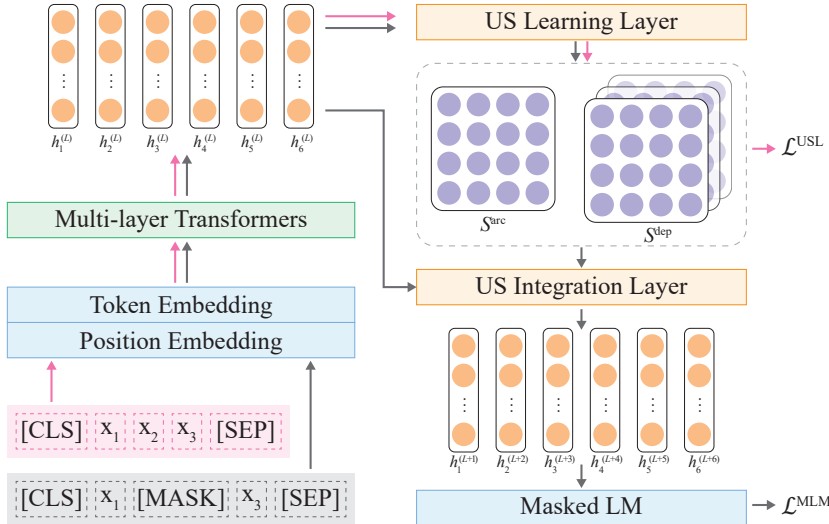

Figure 1: The model architecture for UD-PrLM.

of the structure will further exert its effectiveness in downstream tasks rather than discarding it in the fine-tuning stage, which increases the adaptability from pre-training to fine-tuning phase.

# 3 Universal Dependency as Language Modeling

In this work, we chose UD parse as our universal structure knowledge. Our model includes five modules: token representation, multi-layer Transformers, universal structure learning (USL) layer, universal structure integration (USI) layer and pre-training objectives. Figure 1 shows the full model architecture of our method.

## 3.1 Token Representation

Take BERT [1] as an example, in the token representation layer, given input sentence $X$, the sentence is concatenated with two special tokens "[CLS]" and "[SEP]": [CLS], $x_1, x_2, ..., x_N$, [SEP], and [CLS] is also used as the dummy ROOT node in UD training process. The input $X$ is mapped into a sequence of input embedding vectors $[e_1, e_2, ..., e_{|X|}]$, one for each token, which is a sum of the corresponding word and positional embeddings.

Since the UD parse tree is annotated at word-level and the input sequence $X$ in PrLM is based on subword tokenization, in order to sufficiently improve the representation of all tokens and adapt to the subword-level tasks, we propose a conversion strategy that extends the parsing tree from the word level: $\hat{Y}$ to the subword level: $Y$, and use the subword-level parsing tree as the training objective for USL. Detailed strategy description is shown in Appendix A.1.

## 3.2 Multi-layer Transformers

The multi-layer Transformers architecture in our model is adapted from Vaswani et al.[28], which transforms the input embedding vectors into a sequence of contextualized representation vectors $\mathbf{H} = [h_1, h_2, ..., h_{|X|}]$ shared across different tasks. We use a Transformer architecture with $L$ layers, $A$ self-attention heads for each block and hidden size $H$:

$$\mathbf{H}^{(L)} = \text{Transformers}(\mathbf{Emb}(X) + \mathbf{PosEncoding}(X))$$

**Algorithm 1:** Training Process

---

**Input:** MLM training data $\hat{X}_{MLM}$, UD Treebanks $(\hat{X}_{UD}, \hat{Y}_{UD})$, Parameters: $\theta = (\rho, \gamma, \omega, \phi)$,
      Probability of training USL: $p$.

1   $D_{MLM}, X_{UD}, Y_{UD} \leftarrow \mathbf{Token}(\hat{X}_{MLM} \cup \hat{X}_{UD}), \mathbf{Token}(\hat{X}_{UD}), \mathbf{Strategy}(\hat{Y}_{UD}, \mathbf{Token})$ ;
2   **Initialize** $\theta_0$ randomly ;
3   **for** $t \leftarrow 1$ **to** $m$ **do**
4      $\theta_{t+1} \leftarrow \mathbf{Opt}(\theta_t, \mathcal{L}^{USL}(\rho_t + \gamma_t, X_{UD}^t, Y_{UD}^t) + \mathcal{L}^{MLM}(\rho_t + \phi_t, D_{MLM}^t))$ ;
5   **for** $t \leftarrow m$ **to** $m + n$ **do**
6      **if** random.uniform$(0, 1) < p$ **then**
7          $\theta_{t+1} \leftarrow \mathbf{Opt}(\theta_t, \mathcal{L}^{USL}(\rho_t + \gamma_t, X_{UD}^t, Y_{UD}^t) + \mathcal{L}^{MLM}(\rho_t + \gamma_t + \omega_t + \phi_t, D_{MLM}^t))$ ;
8      **else**
9          $\theta_{t+1} \leftarrow \mathbf{Opt}(\theta_t, \mathcal{L}^{MLM}(\rho_t + \gamma_t + \omega_t + \phi_t, D_{MLM}^t))$;

**Output:** $\theta_n$.

---

### 3.3 Universal Structure Learning

Our USL layer follows the state-of-the-art graph-based deep biaffine dependency parser [29]. We replace the BiLSTM encoder with the multi-layer Transformers architecture and use the hidden state of its last layer as the output of encoder $\mathbf{H}^{(L)} = [h_1^{(L)}, h_2^{(L)}, ..., h_{|X|}^{(L)}]$.

For both arc and label predictions, two separate MLPs are used to distinguish two kinds of low-dimensional vectors as head and dependent representations respectively.

$$r_i^m = \mathbf{ReLU}(\mathbf{MLP}^m(h_i^{(L)})), m \in [head, dep], i = 1, 2, ..., |X|$$

The scores of all possible head-dependent pairs for arc and all head-dependent-label triples for label are computed via the Variable-class biaffine classifier [29]:

$$R_m = [r_1^m; r_2^m; ...; r_{|X|}^m], m \in [head, dep]$$
$$S^k = \mathbf{Softmax}(R_{dep}^T U_1 R_{head} + u_2^T R_{head} + u_3^T R_{dep} + b), k \in [arc, label].$$

For arc, $U_1 \in \mathcal{R}^{H_{dep} \times H_{head}}, u_2 \in \mathcal{R}^{H_{head}}, u_3 \in \mathcal{R}^{H_{dep}}$. For label, $U_1 \in \mathcal{R}^{|D| \times H_{dep} \times H_{head}}, u_2 \in \mathcal{R}^{|D| \times H_{head}}, u_3 \in \mathcal{R}^{|D| \times H_{dep}}$ where $H_{head}$ is the dimension of the head representations, $H_{dep}$ is the dimension of the dependent representations and $D$ is the label set. So that $S^{arc} \in \mathcal{R}^{|X|_{dep} \times |X|_{head}}$ and $S^{label} \in \mathcal{R}^{|D| \times |X|_{dep} \times |X|_{head}}$.

During training, we aim to optimize the following probability for UD parsing:

$$P_\theta(Y|X) = \prod_{i=1}^{|X|} P_\theta(y_i^{label}|x_i, y_i^{arc})P_\theta(y_i^{arc}|x_i),$$

where $\theta$ denotes the learnable parameters and $y_i^{arc}, y_i^{label}$ denote the gold-standard head and dependency relation for subword $x_i$ in subword-level parsing tree $Y$. The training objective for UD parsing is the cross-entropy, which minimizes the negative log-likelihood:

$$\mathcal{L}^{USL} = -\sum_{i=1}^{|X|} \left( \log P_\theta(y_i^{arc}|x_i) + \log P_\theta(y_i^{label}|x_i, y_i^{arc}) \right).$$

For evaluation, we restore the subword-level score tensors: $S^{arc}$ and $S^{label}$ to word-level: $\hat{S}^{arc}$ and $\hat{S}^{label}$ by extracting the first subword of each word. Then, we judge whether the prediction result of $\hat{S}^{arc}$ is a valid parsing tree. If so, we directly extract the corresponding prediction label from $\hat{S}^{label}$. Otherwise, we use the max spanning tree (MST) algorithm to find the maximum spanning tree based on $\hat{S}^{arc}$.

## 3.4 Universal Structure Integration

In order to better integrate linguistic structure knowledge into the output representation of our PrLM, we propose the USI layer, which combines $S^{arc}$ and $S^{label}$ obtained by the USL layer with $\mathbf{H}^{(L)}$ as the final output representations.

We first combine $S^{arc}$ and $S^{label}$ into a full label scoring matrix $S^{US}$ by dot product. $S^{US} \in \mathcal{R}^{|X|_{dep} \times |X|_{head} \times |D|}$, in fact, stores the information about the label-head pair probability of each dependent in the sentence. Then we use this label scoring matrix $S^{US}$ as the attention score to obtain a dependent and label specific representation by product summation operation to $\mathbf{H}^{(L)}$, the result is $\hat{\mathbf{H}}^{US} \in \mathcal{R}^{|X| \times H \times |D|}$.

$$S^{US} = S^{arc} \cdot S^{label}, \quad \hat{\mathbf{H}}^{US} := S^{US}_{ijk} \times \mathbf{H}^{(L)}_{ih} \to O_{ihk},$$

where $[\cdot]_{ijk} \times [\cdot]_{ih} \to [\cdot]_{ihk}$ indicates the Einstein summation notation.

Then we employ a weight tensor $W \in \mathcal{R}^{|D| \times H \times H}$ to aggregate and map the dependent and label specific representation to the final dependency tree-aware representation: $\mathbf{H}^{US} \in \mathcal{R}^{|X| \times H}$.

$$\mathbf{H}^{O} := \hat{\mathbf{H}}^{US}_{ihk} \times W_{khm} \to O_{ihk}, \quad \mathbf{H}^{US} = \mathbf{GELU}(\mathbf{Linear}(\mathbf{H}^{O})),$$

where $[\cdot]_{ihk} \times [\cdot]_{khm} \to [\cdot]_{ihk}$ indicates the Einstein summation notation, the dimensions of $\mathbf{H}^{O} \in \mathcal{R}^{|X| \times H \times |D|}$, and will be flatten to shape $\mathcal{R}^{|X| \times (H \times |D|)}$ before input to $\mathbf{Linear}$.

We also do a residual connection to avoid losing the information in $\mathbf{H}^{(L)}$, and we use an additional Transformer layer to get the final representation.

$$\mathbf{H}^{(L+1)} = \mathbf{Transformer}(\mathbf{H}^{US} + \mathbf{H}^{(L)})$$

## 3.5 Pre-training Objectives

We use Masked LM (MLM) as the only pre-training objective other than USL. In MLM, a random sample of the tokens in the input sequence is selected and replaced with the special token [MASK]. As described in BERT [1], 15% of the input tokens are uniformly selected for possible replacement. Of the selected tokens, 80% are replaced with [MASK], 10% are left unchanged, and 10% are replaced by a randomly selected vocabulary token. The MLM objective is a cross-entropy loss on predicting the masked tokens:

$$\mathcal{L}^{MLM} = - \sum_{x \in m(X)} log P_{\theta}(x | X_{\backslash m(X)}),$$

where $m(X)$ and $X_{\backslash m(X)}$ denote the masked words from $X$ and the rest of words respectively. The training loss of our model is calculated by adding the losses of USL and MLM.

$$\mathcal{L} = \mathcal{L}^{USL} + \mathcal{L}^{MLM}$$

## 3.6 Training Details

Algorithm 1 shows the training process of our model, in which **Token** denotes the tokenization, **Strategy** denotes the strategy creating subword-level parsing tree, **Opt** denotes the optimization strategy and $\rho, \gamma, \omega, \phi$ denote the learnable parameters in Transformer encoder (and token representation), USL layer, USI layer and MLM decoder respectively. We randomly initialize the model parameters $\theta$. In the first $m$ epochs, USL and MLM are trained as two parallel tasks sharing the parameters in Transformer encoder. That is to say, we do not use USI layer to integrate universal structure knowledge for MLM because the parsing capability of the model is too weak in this phase. In the next $n$ steps, we open the USI layer. At this time, the USL objective has converged well, so we reduce the frequency of training USL appropriately according to a certain probability $p$.

# 4 Experiments

## 4.1 Setup

**Pre-training Data**    Similar to that of m-BERT, we chose the top 104 languages with the largest Wikipedias and apply exponentially smoothed weighting to these languages to balance the Wikipedia

size of each language for training MLM in the resulted UD-BERT according to our proposed approach. We do not use the NSP objective for UD-BERT. For UD-XLM-R$_{base}$ and UD-XLM-R$_{large}$, we use the same training set from CommonCrawl Corpus as in Conneau et al.(2019) [18]. We also used MLM as the only objective other than USL for UD-XLM-R$_{base}$ and UD-XLM-R$_{large}$. For structure learning, we concatenate all the training TreeBanks covering 60 languages in Universal Dependencies Treebanks (v2.2) [30] as the training set. In addition, we add the sentences in the training TreeBanks of UD to the training set of MLM letting language modeling directly help our model learn the structure knowledge.

**Pre-training Settings** In order to be consistent with the baselines and minimize the inequity of the amount of parameters, we use the same settings as the baselines for $H$, $A$ and vocabulary. However, for the number of Transformer layers, since we use an additional Transformer layer in USI, the number of layers $L$ prior to this should be reduced by one from the baselines. Specifically, our UD-BERT and UD-XLM-R$_{base}$ use a Transformer architecture with $L = 11, H = 768$ and $A = 12$ with a vocabulary of 110k and 250k respectively. Our UD-XLM-R$_{large}$ uses a large Transformer architecture with $L = 23, H = 1024$ and $A = 16$ with a 250k vocabulary. We use WordPiece [31] tokenization of UD-BERT and SentencePiece [32] tokenization for UD-XLM-R$_{base}$ and UD-XLM-R$_{large}$. We randomly initialize the model parameters rather than using the pre-trained parameters of m-BERT or XLM-R for fair comparison with the baselines, so as to avoid the improvement of training from more training steps. We train our models with the Adam optimizer [33] using the parameters: Learning rate $= 5e - 5, \beta_1 = 0.9, \beta_2 = 0.98, \epsilon = 1e - 6$ and $L_2$ weight decay of 0.01, a linear warmup [28], GELU activation [34] and a dropout rate of 0.1. Models are trained for $m = 600,000$ and $n = 600,000$ epochs in each phase respectively, with Batch size $= 128$(sents), and the probability of training USL in the second phase is $p = 0.8$. The max sequence length of MLM is 384 and the max sequence length for UD parsing is 256. In the USL layer, we set $H_{head} = 128$ and $H_{dep} = 64$. We list the parameters of our full models and baselines, as well as the data statistics of our training data in Table 7 in Appendix A.2.

**XNLI: Cross-lingual Natural Language Inference** takes two sentences as input and determines whether one entails the other, contradicts it or neither. XNLI is defined on 15 languages. Each language contains a development set with 2,490 sentence pairs and a test set with 5,010 sentence pairs. Only English has training data, which is a crowd-sourced collection of 433k sentence pairs from MultiNLI [35]. The performance is evaluated by classification accuracy.

**XQuAD: Cross-lingual Question Answering Dataset** [36] is a benchmark dataset for evaluating cross-lingual question answering performance. The dataset consists of a subset of 240 paragraphs and 1,190 question-answer pairs from the development set of SQuAD v1.1 [6] together with their professional translations into 10 languages. The performance is evaluated by F1 and exact match (EM) scores.

**Universal Linguistic Structure Parsing:** For universal dependency parsing, we evaluate our model on 22 languages in Universal Dependencies Treebanks (v2.7) [30] whose detail information is shown in Table 8 in Appendix A.2. We use the graph-based deep biaffine dependency parsing model [29] as our dependency parser. For monolingual evaluation, we train a model for each language on their training set using word, character and POS tag embeddings of dimension 100 and representation from PrLM of dimension 300. For cross-lingual evaluation, we train a single model on English training set using POS tag and representation from PrLM. Unlabeled Attachment Scores (UAS) and Labeled Attachment Scores (LAS) are adopted as the evaluation metrics. For universal constituent parsing, we explore the improvement of our model using SPMRL Shared Task 2014, which focuses on parsing nine morphologically rich languages from different language families. We also evaluate our model on PTB and CTB. We use the CRF constituency parsing model [37] as our constituent parser with word and character embeddings of dimension 100 and representation from PrLM of dimension 300 for monolingual evaluation. For cross-lingual evaluation, we train a single model on PTB using only the representation from PrLM.

## 4.2 Results and Analysis

To evaluate the multilingual performance and cross-lingual transfer effect of the PrLM that learns the universal linguistic structure and integrates the universal linguistic structure into the represen-

tation explicitly, we conducted experiments on the two typical tasks: universal natural language understanding and the universal linguistic structure parsing. All the scores are average results of five random seeds, indicating that our results are stable.

**Universal Natural Language Understanding** In Table 1, we show the cross-lingual transfer results (*Cross-Transfer*) of the baselines and our proposed model on the cross-lingual text classification benchmark - XNLI. Meanwhile, we also list the multilingual performance (*Train-Trans-FT*, *Test-Trans-Eval*, and *All-FT*) as a reference. First, compare the results of the source language - English, our UD-BERT, UD-XLM-R$_{base}$, and UD-XLM-R$_{large}$ outperform the corresponding m-BERT, XLM-R$_{base}$, and XLM-R$_{large}$ baselines, demonstrating the universal linguistic structure as pre-training objective and explicitly syntactic structure integration improve the model pre-training and final representations.

Table 1: Results on cross-lingual text classification task. We report the accuracy on each of the 15 XNLI languages and the average accuracy. Results with $^{\dagger}$ are from [26].

| Model | en | fr | es | de | el | bg | ru | tr | ar | vi | th | zh | hi | sw | ur | Avg |
|---|---|---|---|---|---|---|---|---|---|---|---|---|---|---|---|---|
| *Train-Trans-FT: Fine-tune multilingual model on each training set translated from English* | | | | | | | | | | | | | | | | |
| XLM [9] | 82.9 | 77.6 | 77.9 | 77.9 | 77.1 | 75.7 | 75.5 | 72.6 | 71.2 | 75.8 | 73.1 | 76.2 | 70.4 | 66.5 | 62.4 | 74.2 |
| *Test-Trans-Eval: Translate test sets to English and use English-only model for evaluation* | | | | | | | | | | | | | | | | |
| BERT-en | 88.8 | 81.4 | 82.3 | 80.1 | 80.3 | 80.9 | 76.2 | 76.0 | 75.4 | 72.0 | 71.9 | 75.6 | 70.0 | 65.8 | 65.8 | 76.2 |
| RoBERTa | 91.3 | 82.9 | 84.3 | 81.2 | 81.7 | 83.1 | 78.3 | 76.8 | 76.6 | 74.2 | 74.1 | 77.5 | 70.9 | 66.7 | 66.8 | 77.8 |
| *All-FT: Fine-tune multilingual model on all training sets* | | | | | | | | | | | | | | | | |
| XLM [9] | 84.5 | 80.1 | 81.3 | 79.3 | 78.6 | 79.4 | 77.5 | 75.2 | 75.6 | 78.3 | 75.7 | 78.3 | 72.1 | 69.2 | 67.7 | 76.9 |
| XLM [9]$^{\dagger}$ | 85.0 | 80.8 | 81.3 | 80.3 | 79.1 | 80.9 | 78.3 | 75.6 | 77.6 | 78.5 | 76.0 | 79.5 | 72.9 | 72.8 | 68.5 | 77.8 |
| Unicoder [26] | 85.6 | 81.1 | 82.3 | 80.9 | 79.5 | 81.4 | 79.7 | 76.8 | 78.2 | 77.9 | 77.1 | 80.5 | 73.4 | 73.8 | 69.6 | 78.5 |
| XLM-R$_{base}$ | 85.4 | 81.4 | 82.2 | 80.3 | 80.4 | 81.3 | 79.7 | 78.6 | 77.3 | 79.7 | 77.9 | 80.2 | 76.1 | 73.1 | 73.0 | 79.1 |
| XLM-R$_{large}$ | 89.1 | 85.1 | 86.6 | 85.7 | 85.3 | 85.9 | 83.5 | 83.2 | 83.1 | 83.7 | 81.5 | 83.7 | 81.6 | 78.0 | 78.1 | 83.6 |
| *Cross-Transfer: Fine-tune multilingual model on English training set* | | | | | | | | | | | | | | | | |
| XLM [9] | 85.0 | 78.7 | 78.9 | 77.8 | 76.6 | 77.4 | 75.3 | 72.5 | 73.1 | 76.1 | 73.2 | 76.5 | 69.6 | 68.4 | 67.3 | 75.1 |
| Unicoder [26] | 85.1 | 79.0 | 79.4 | 77.8 | 77.2 | 77.2 | 76.3 | 72.8 | 73.5 | 76.4 | 73.6 | 76.2 | 69.4 | 69.7 | 66.7 | 75.4 |
| m-BERT [1] | 82.1 | 73.8 | 74.3 | 71.1 | 66.4 | 68.9 | 69.0 | 61.6 | 64.9 | 69.5 | 55.8 | 69.3 | 60.0 | 50.4 | 58.0 | 66.3 |
| **UD-BERT** | **82.7** | **74.9** | **75.2** | **72.0** | **67.4** | **69.2** | **70.3** | **62.7** | **65.8** | **70.3** | **59.6** | **69.7** | **61.4** | **51.2** | **58.7** | **67.4** |
| XLM-R$_{base}$ | 85.8 | 79.7 | 80.7 | 78.7 | 77.5 | 79.6 | 78.1 | 74.2 | 73.8 | 76.5 | 74.6 | 76.7 | 72.4 | 66.5 | 68.3 | 76.2 |
| **UD-XLM-R$_{base}$** | **86.5** | **80.3** | **81.6** | **79.8** | **78.4** | **80.0** | **78.9** | **75.1** | **74.4** | **77.1** | **75.2** | **77.3** | **73.0** | **67.0** | **68.8** | **76.9** |
| XLM-R$_{large}$ | 89.1 | 84.1 | 85.1 | 83.9 | 82.9 | 84.0 | 81.2 | 79.6 | 79.8 | 80.8 | 78.1 | 80.2 | 76.9 | 73.9 | 73.8 | 80.9 |
| **UD-XLM-R$_{large}$** | **89.4** | **84.8** | **85.6** | **84.5** | **83.6** | **84.7** | **81.2** | **80.0** | **81.0** | **81.9** | **78.6** | **80.7** | **76.8** | **74.4** | **74.3** | **81.4** |

Second, our UD-BERT and UD-XLM-R performed better in most cases of the 14 transferring target languages. UD-BERT has an average increase of 1.1 when compared to the baseline, UD-XLM-R$_{base}$ has an average increase of 0.7, and UD-XLM-R$_{large}$ has an average increase of 0.5. This improvement highlights the fact that the cross-lingual transferring ability of our multilingual PrLM has improved as a result of the employment of universal linguistic structures. Furthermore, the cross-lingual transfer effect of our UD-XLM-R$_{base}$ outperforms BERT with a similar model structure and parameters, who translated the test set for evaluation. The results of our UD-XLM-R$_{large}$ achieved better results than all the methods which leveraging a monolingual language model on the translations, which shows that cross-lingual transfer is a more promising mode.

Text classification is a relatively simple and intuitive NLU task. To further verify our conclusions, we conducted experiments on a more complex task - cross-lingual Machine Reading Comprehension (MRC). The results on XQuAD dataset are shown in Table 2. Similarly, we first compare the MRC results on the source language English. The performance of UD-BERT and UD-XLM-R has increased relative to the baseline, which verifies the conclusion that our universal linguistic structure improves the NLU ability of multilingual language model. For the 11 transferring target languages, we observed a similar improvement trend on UD-BERT and UD-XLM-R as in the XNLI task, and the improvement was even greater. The average improvement of UD-BERT, UD-XLM-R base and UD-XLM-R large reached 1.0, 0.6, 0.8 F1 scores respectively. Among them, Arabic has the largest improvement, with 1.3, 2.3, and 6.6 F$_1$ scores respectively. All the results here reveal that universal syntactic structure information embedded is effective for cross-lingual MRC task.

**Universal Linguistic Structure Parsing** Since the universal dependency parsing structure and the dependency parse encoding structure are built into our multilingual PrLM, to demonstrate that our model learned how to extract linguistic structure and to encode the structure into the output

Table 2: $F_1$ / EM scores on XQuAD with English as the source language for each target language.

| Cross-Transfer | | en | ar | de | el | es | hi | ru | th | tr | vi | zh | ro | Avg |
|---|---|---|---|---|---|---|---|---|---|---|---|---|---|---|
| m-BERT | $F_1$ | 83.5 | 61.5 | 70.6 | 62.6 | 75.5 | 59.2 | 71.3 | 42.7 | 55.4 | 69.5 | 58.0 | 72.7 | 65.2 |
| | EM | 72.2 | 45.1 | 54.0 | 44.9 | 56.9 | 46.0 | 53.3 | 33.5 | 40.1 | 49.6 | 48.3 | 59.9 | 50.3 |
| **UD-BERT** | $F_1$ | **83.9** | **62.8** | **72.3** | **62.9** | **75.7** | 59.0 | **71.6** | **48.6** | **55.5** | **69.9** | **58.7** | **73.7** | **66.2** |
| | EM | **72.5** | **46.3** | **56.0** | **45.8** | **57.3** | **46.3** | **54.0** | **38.5** | **40.2** | **49.7** | **48.6** | **60.3** | **51.3** |
| XLM-R$_{base}$ | $F_1$ | 83.6 | 66.8 | 74.4 | 73.0 | 76.4 | 68.2 | 74.3 | 66.5 | 68.3 | 73.7 | 51.3 | 77.8 | 71.2 |
| | EM | 72.1 | 49.1 | 60.1 | 55.7 | 58.3 | 51.7 | 58.1 | 56.7 | 52.8 | 53.8 | 42.0 | 62.8 | 56.1 |
| **UD-XLM-R$_{base}$** | $F_1$ | **84.0** | **69.1** | **74.9** | **73.5** | **77.0** | **68.4** | 74.3 | **66.9** | **69.3** | **74.1** | **51.8** | **78.0** | **71.8** |
| | EM | **72.5** | **51.2** | **60.5** | **56.0** | 58.3 | **51.9** | **58.5** | **57.1** | **52.9** | **54.2** | **42.6** | **63.3** | **56.6** |
| XLM-R$_{large}$ | $F_1$ | 86.5 | 68.6 | 80.4 | 79.8 | 82.0 | 76.7 | 80.1 | 74.2 | 75.9 | 79.1 | 59.3 | 83.6 | 77.2 |
| | EM | 75.7 | 49.0 | 63.4 | 61.7 | 63.9 | 59.7 | 64.3 | 62.8 | 59.3 | 59.0 | 50.0 | 69.7 | 61.5 |
| **UD-XLM-R$_{large}$** | $F_1$ | **86.8** | **75.2** | **80.9** | **80.0** | **82.3** | **77.1** | **80.3** | 73.8 | **76.3** | **79.5** | **59.4** | **83.9** | **78.0** |
| | EM | **75.9** | **58.2** | **63.8** | 61.7 | **64.0** | **59.8** | **64.5** | 62.2 | **60.5** | **59.8** | 49.9 | **69.9** | **62.5** |

Table 3: The monolingual UD parsing results (UAS/LAS) on 22 UD Treebanks. $^*$ means that the PrLM is tested directly without additional training of a biaffine dependency parsing model.

| All-FT | m-BERT | | **UD-BERT**$^*$ | | **UD-BERT** | | XLM-R$_{base}$ | | **UD-XLM-R$_{base}$** | | XLM-R$_{large}$ | | **UD-XLM-R$_{large}$** | |
|---|---|---|---|---|---|---|---|---|---|---|---|---|---|---|
| | UAS | LAS | UAS | LAS | UAS | LAS | UAS | LAS | UAS | LAS | UAS | LAS | UAS | LAS |
| **bg** | 94.75 | 90.88 | 95.76 | 91.57 | **96.12** | **93.09** | 96.42 | 93.56 | **96.57** | **93.80** | 96.53 | 93.72 | **96.69** | **94.04** |
| **ca** | 95.36 | 93.25 | 94.97 | 92.72 | **95.61** | **94.27** | 95.53 | 94.26 | **95.78** | **94.61** | 95.75 | 94.58 | **95.94** | **94.72** |
| **cs** | 94.38 | 91.62 | 95.11 | 91.94 | **95.62** | **93.17** | 95.60 | 93.30 | **95.82** | **93.43** | 95.87 | 93.69 | **95.99** | **93.87** |
| **nl** | 94.74 | 92.72 | 94.44 | 91.27 | **95.38** | **93.61** | 95.36 | 93.44 | **95.79** | **93.82** | 95.63 | 93.78 | **96.13** | **94.50** |
| **en** | 92.52 | 91.29 | 91.34 | 88.17 | **93.01** | **91.43** | 93.60 | 91.83 | **94.15** | **92.55** | 93.47 | 91.73 | **94.19** | **92.54** |
| **et** | 90.88 | 88.95 | 90.04 | 86.23 | **91.65** | **89.73** | 92.53 | 90.78 | **92.78** | **91.02** | 93.16 | 91.50 | **93.28** | **91.68** |
| **fi** | 92.98 | 90.65 | 89.41 | 82.51 | **94.07** | **91.89** | 94.99 | 93.14 | **95.15** | **93.44** | 95.17 | 93.46 | **95.66** | **94.01** |
| **fr** | 94.12 | 90.75 | 95.52 | 92.20 | **96.08** | **94.24** | 96.10 | 94.34 | **94.69** | **94.69** | 96.01 | 94.15 | **96.53** | **94.75** |
| **de** | 90.77 | 86.83 | 88.64 | 81.78 | **91.08** | **87.35** | 91.30 | 87.42 | **91.58** | **87.63** | 91.39 | 87.50 | **91.72** | **87.94** |
| **he** | 92.32 | 89.95 | 92.52 | 88.55 | **93.45** | **91.06** | 93.50 | 91.41 | **93.87** | **91.64** | 93.67 | 91.48 | **93.99** | **91.79** |
| **hi** | 96.54 | 94.22 | 95.25 | 91.34 | **96.71** | **94.41** | 96.73 | 94.51 | **97.03** | **94.99** | 96.93 | 94.76 | **97.17** | **95.09** |
| **id** | 88.29 | 83.97 | 87.73 | 79.18 | 87.96 | 83.72 | 88.25 | 84.06 | **88.51** | **84.25** | 88.38 | 84.27 | **88.48** | 84.19 |
| **it** | 95.63 | 94.01 | 95.99 | 93.59 | **96.32** | **95.16** | 96.15 | 94.93 | **96.70** | **95.33** | 96.26 | 95.01 | **96.82** | **95.60** |
| **ko** | 90.73 | 88.27 | 81.90 | 71.29 | **90.99** | **88.67** | 91.33 | 89.25 | **91.42** | **89.30** | 92.15 | 89.79 | **92.16** | **89.98** |
| **la** | 85.69 | 81.84 | 84.57 | 79.04 | **87.01** | **83.45** | 86.64 | 82.99 | **89.86** | **86.91** | 86.97 | 83.29 | **90.44** | **87.33** |
| **lv** | 91.55 | 89.06 | 90.57 | 85.65 | **92.06** | **89.72** | 93.53 | 91.25 | **93.88** | **91.50** | 94.23 | 91.90 | **94.56** | **92.52** |
| **no** | 95.62 | 93.88 | 94.61 | 92.64 | **96.23** | **95.17** | 96.57 | 95.58 | **96.71** | **95.63** | 96.70 | 95.64 | **96.74** | **95.75** |
| **pl** | 98.15 | 96.54 | 96.26 | 88.10 | **98.54** | **97.14** | 98.51 | 97.22 | **98.80** | **97.77** | 98.47 | 97.08 | **98.79** | **97.74** |
| **ro** | 92.70 | 86.39 | 93.58 | 87.38 | **94.12** | **89.39** | 94.25 | 89.67 | **94.60** | **90.11** | 94.39 | 89.85 | **94.73** | **90.32** |
| **ru** | 95.26 | 94.00 | 95.30 | 92.95 | **95.74** | **94.58** | 96.25 | 95.26 | **96.54** | **95.59** | 96.38 | 95.45 | **96.62** | **95.72** |
| **sk** | 94.93 | 91.40 | 96.22 | 93.19 | **96.62** | **93.95** | 95.96 | 93.21 | **97.49** | **95.53** | 95.56 | 92.74 | **97.26** | **95.34** |
| **es** | 94.69 | 92.89 | 94.08 | 90.72 | **94.92** | **93.23** | 94.98 | 93.44 | **95.38** | **93.80** | 95.33 | 93.80 | **95.55** | **94.09** |
| **Avg** | 93.30 | 90.61 | 92.45 | 87.82 | **94.06** | **91.75** | 94.28 | 92.04 | **94.77** | **92.61** | 94.47 | 92.24 | **94.97** | **92.89** |

representations, we performed the universal linguistic structure parsing on UD and multilingual constituent parsing tasks. It is worth noting that, to shield the influence of the increase in the number of parameters caused by the addition of the PrLM to the downstream task model, we kept all the PrLM parameters frozen in the universal NLU evaluation.

In Table 3, we evaluated the effect of using the multilingual PrLM to enhance the parsing model on 22 languages of UD dataset respectively, in order to analyze how many features useful (i.e., syntactic-aware features) for parsing are provided by the output representation of the various multilingual PrLM. At the same time, to verify that our model has learned syntax, we listed the results that UD-BERT is tested directly on 22 languages without additional training of a biaffine dependency model.

Comparing UD-BERT and m-BERT, our UD-BERT obtained generally better results, except for Indonesian, with a 0.76/1.14 UAS/LAS average improvement. The average score of UD-BERT$^*$ is comparable to that of m-BERT, and it outperforms m-BERT in some languages, indicating that our model has integrated a well-performing parser that can be directly applied to practice without additional training. When we further compare UD-BERT to stronger XLM-R$_{base}$ and XLM-R$_{large}$ baselines, we found that XLM-R$_{base}$ achieved similar results as UD-BERT. Since XLM-R uses more data and longer pre-training time, from this point of view, UD-BERT has played a role in reducing

Table 4: Labeled $F_1$ scores on PTB, CTB, and SPMRL test sets.

| All-FT | en | ar | eu | fr | de | he | hu | ko | pl | sv | zh | Avg |
|---|---|---|---|---|---|---|---|---|---|---|---|---|
| m-BERT | 94.87 | 88.60 | 91.06 | 84.98 | 90.20 | 83.35 | 92.33 | 89.36 | 96.36 | 83.05 | 90.83 | 89.54 |
| **UD-BERT** | **95.09** | **89.97** | **92.23** | **85.57** | **91.06** | **84.36** | **93.46** | **89.72** | **96.59** | **85.06** | **90.99** | **90.37** |
| XLM-R$_{base}$ | 95.50 | 89.67 | 91.68 | 85.63 | 91.54 | 85.20 | 93.92 | 90.82 | 96.72 | 86.00 | 91.84 | 90.77 |
| **UD-XLM-R$_{base}$** | **95.83** | **90.22** | **92.67** | **86.10** | **91.98** | **86.03** | **94.61** | **91.32** | **96.95** | **86.71** | **92.30** | **91.34** |
| XLM-R$_{large}$ | 95.74 | 90.41 | 92.54 | 86.47 | 92.10 | 86.11 | 94.56 | 91.77 | 97.15 | 86.80 | 92.24 | 91.44 |
| **UD-XLM-R$_{large}$** | **96.15** | **90.77** | **94.13** | **86.96** | **92.65** | **86.97** | **95.41** | **92.16** | **97.33** | **89.06** | **92.43** | **92.18** |

Table 5: Unlabeled $F_1$ scores on PTB, CTB, and SPMRL test sets.

| Cross-Transfer | en | ar | eu | fr | de | he | hu | ko | pl | sv | zh | Avg |
|---|---|---|---|---|---|---|---|---|---|---|---|---|
| m-BERT | 95.54 | 24.22 | 32.61 | 59.40 | 44.36 | 46.01 | 57.86 | 33.99 | 37.47 | 71.69 | 56.87 | 50.91 |
| **UD-BERT** | **95.93** | **27.09** | **35.50** | **60.87** | **45.24** | **46.88** | **58.99** | **35.67** | **39.23** | **72.20** | **57.74** | **52.30** |
| XLM-R$_{base}$ | 96.19 | 28.17 | 34.34 | 59.70 | 44.98 | 47.05 | 60.26 | 38.95 | 38.06 | 73.68 | 55.36 | 52.43 |
| **UD-XLM-R$_{base}$** | **96.60** | **30.46** | **37.75** | **61.36** | **45.52** | **47.42** | **62.00** | **42.94** | **39.83** | **73.90** | **59.97** | **54.34** |
| XLM-R$_{large}$ | 96.44 | 20.56 | 34.23 | 59.81 | 45.04 | 47.01 | 59.37 | 36.72 | 37.87 | 73.43 | 54.34 | 51.35 |
| **UD-XLM-R$_{large}$** | **96.73** | **23.44** | **37.63** | **61.18** | **45.71** | **47.30** | **60.46** | **41.26** | **38.42** | **73.81** | **59.58** | **53.23** |

the data and time required for pre-training due to the addition of syntactic supervision information. Further comparing UD-XLM-R with the baseline XLM-R, we found that our method is still helpful on such strong baselines, which improves the parsing performance in each language. In addition, we also evaluate the improvement of our model in zero-shot cross-lingual[3] setting and in low-resource languages. The results are shown in Appendices A.3 and A.4, respectively.

Though we verified the linguistic feature extraction and encoding capabilities of our PrLM in the UD parsing, since the evaluation task is the same as in the pre-training, it is hard to illustrate this effect on the cross-task linguistic parsing task. Thus, we conduct further exploration on multilingual constituent parsing benchmarks. Table 4 shows the enhanced ability of the language model for monolingual constituent parsing. The comparison demonstrates that with the help of additional universal dependency features in the representations, the constituent parsing performance of UD-BERT and UD-XLM-R has been greatly increased, and the average improvement has reached 0.83, 0.57, and 0.74 respectively. This reflects that the universal syntactic features contained in the representations are very helpful for downstream tasks. In addition, with our syntax-aware multilingual PrLM, the integration of syntactic tree is no longer necessary to change the downstream task model for adding extra syntactic encoders, which will greatly reduce the cost of syntactic tree information application.

The multilingual boosting ability of our PrLM is shown in Table 4. Similarly, we evaluated the PrLM's cross-lingual transfer ability on constituent parsing in Table 5, and because there are no available universal constituent label annotations, we only report unlabeled F1. Again, our UD-BERT and UD-XLM-R achieve an improvement in all transferring target languages, which shows that the explicit universal features contained in our PrLM can be used to help downstream cross-lingual transfer tasks. Since the universal dependency structure can be derived directly by additional inference in our PrLM's intermediate output, our good cross-lingual transfer ability is interpretable. Furthermore, our studies found that the transfer effect of XLM-R$_{base}$ is superior to that of XLM-R$_{large}$, implying that larger model parameters may not always result in higher cross-lingual transfer capabilities.

## 5 Ablation Study

In our multilingual PrLM, we propose both novel model structure and novel pre-training strategy, whose effectiveness is verified by the ablation study in Table 6. We also doubt that the gain is partly derived from the additional UD plain text, although the amount of UD plain text is small. So we also show the results of training the baseline with additional UD plain text (*w/ UD plain text*) in Table 6. From the results, if universal dependency parsing is only used as an additional pre-training process (*w/o US Integration*), there is only a slight performance improvement compared to the baseline,

---

[3]In the strict sense, this cannot be totally categorized as zero-shot because the language model part of the full parser has been pre-trained by UD annotations.

which shows that just as a pre-training approach is not enough to fuse the universal syntax implicitly. When we only employ the model structure without UD pre-training (*w/o UD Pre-training*), this additional parser structure, even though it is not trained by the UD supervision, can still improve the performance, and the improvement is greater than that of *w/o US Integration*. Comparing m-BERT and m-BERT(*w/ UD plain text*), we find that the performance difference between them is very small, and the performance is even lower after the addition of UD plain text. This indicates that a small amount of UD plain text can not directly improve the model. Combining the above results with that of our full UD-BERT, we can conclude that the supervised learning of the UD structure and its explicit integration into the representation are indispensable options for performance improvement. And our universal structure integration can help the model learn the syntactic knowledge without additional supervision and improve the performance because it has an accountable form that can well model the syntactic structure features.

Table 6: Ablation studies on XQuAD and SPMRL.

| Method | XQuAD(*Cross-Transfer*) | | | | | | | | SPMRL(*All-FT*) | | | |
| | ar | | de | | en | | el | | ar | de | en | eu |
| | F$_1$ | EM | F$_1$ | EM | F$_1$ | EM | F$_1$ | EM | F$_1$ | F$_1$ | F$_1$ | F$_1$ |
| m-BERT | 61.5 | 45.1 | 70.6 | 54.0 | 83.5 | 72.2 | 62.6 | 44.9 | 88.6 | 90.2 | 94.9 | 91.1 |
| *w/ UD plain text* | 61.0 | 44.6 | 70.1 | 53.8 | 82.9 | 71.8 | 62.3 | 44.8 | 88.5 | 90.2 | 94.6 | 91.0 |
| **UD-BERT** | 62.8 | 46.3 | 72.3 | 56.0 | 83.9 | 72.5 | 62.9 | 45.8 | 90.0 | 91.1 | 95.1 | 92.2 |
| *w/o US Integration* | 61.9 | 45.4 | 70.9 | 54.5 | 83.8 | 72.6 | 62.5 | 45.0 | 88.9 | 90.4 | 94.8 | 91.3 |
| *w/o UD Pre-training* | 61.5 | 45.0 | 70.7 | 54.3 | 83.6 | 72.2 | 62.7 | 45.2 | 89.1 | 90.6 | 94.9 | 91.5 |

# 6 Conclusion

In this work, we propose a multilingual PrLM that supports both explicit universal structure learning and implicit language modeling. We chose the universal dependency parses as our universal structure knowledge and evaluate the cross-lingual modeling capability of our model on two cross-lingual NLU tasks and four syntactic parsing datasets. Our model outperforms m-BERT and XLM-R in all tasks and achieves state-of-the-art results on syntactic parsing. Unlike other works that use the syntax as an objective or feature, our structure learning is used not only for a pre-training objective but also for improving the representation, which makes our model both a PrLM and a universal dependency parser. This will greatly change the way that downstream NLP tasks use syntax, because we have explicitly integrated the syntactic knowledge into the representation of the PrLM.

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
