

Figure 2: An example of extending a word-level parsing tree to a subword-level.

Table 7: Model parameters and training data where CC denotes CommonCrawl.

|  | UD-BERT | m-BERT | UD-XLM-R$_{base}$ | XLM-R$_{base}$ | UD-XLM-R$_{large}$ | XLM-R$_{large}$ |
|---|---|---|---|---|---|---|
| parameters | 192.0M | 177.9M | 291.8M | 278.0M | 581.1M | 559.9M |
| training data | UD+Wiki | Wiki | UD+CC | CC | UD+CC | CC |
|  | (373.16M) | (372.10M) | (2502.26M) | (2501.20M) | (2502.26M) | (2501.20M) |

# A   Appendix

## A.1   Extending Strategy for Parsing Treebanks

Denote a sentence as $\hat{X} = [w_1, w_2, ..., w_N]$ and its corresponding subword-level sentence as $X = [[CLS], w_{11}, ..., w_{1s_1}, w_{21}, ..., w_{2s_2}, ..., w_{N1}, ..., w_{Ns_N}, [SEP]]$. In the word-level parsing tree, for $w_i \in \hat{X}$, suppose the head of $w_i$ is $w_{h_i}$ and the dependency label is $l_i$. We design three strategies to extend the parsing tree from word level to subword level. All of them regard [CLS] as the ROOT ($w_{01}$), and the header of [SEP] as [CLS], and the label as the padding label "_".

- Set the head of all the subwords of $w_i$ as $w_{h_i 1}$, and the label as $l_i$.
- Set the head of $w_{i1}$ as $w_{h_i 1}$ and the label as $l_i$, and set the head of $w_{i2}, ..., w_{is_i}$ as $w_{i1}$ and label as a special label "APP".
- Set the head of $w_{i1}$ as $w_{h_i 1}$ and the label as $l_i$, and set the head of $w_{ij}$ as $w_{i(j-1)}$ for $j = 2, ..., s_i$ and label as a special label "APP". As shown in Figure 2.

## A.2   Details of Experimental Data and Model Parameters

## A.3   Cross-lingual Results on UD Treebanks

Table 9 shows the zero-shot cross-lingual transfer experiments of our PrLM on the 22 languages of UD Treebanks. The findings from comparison reveal that our UD-BERT and UD-XLM-R have significantly improved the transfer effect of this parser in which the average improvement of UD-BERT and UD-XLM-R is 9.93/8.09, 8.86/8.32 UAS/LAS, respectively. This is due in large part to the usage of UD annotations in the language model pre-training, but it also demonstrates that our PrLM well learned UD parsing and encoded the structural information of the parse into the final representations.

## A.4   Monolingual Results on Low-resource Treebanks in UD

We select all treebanks in UD with training sets under 100 sentences to evaluate the performance of our model in low-resource languages. The results are shown in Table 10, where we compare our method with the baselines and the biaffine model without PrLM (*w/o PrLM*). It can be seen that the results on UD-BERT and UD-XLM-R all outperform their baseline. This shows that our method can partially solve the problem of poor performance of the multilingual PrLM in low-resource languages.

Table 8: Details of the selected languages in UD.

| Language | Treebank | Sents |
|---|---|---|
| Bulgarian (bg) | BTB | 8,907 |
| Catalan (ca) | AnCora | 13,123 |
| Czech (cs) | PDT | 102,993 |
| Dutch (nl) | Alpino | 18,058 |
| English (en) | EWT | 12,543 |
| Estonian (et) | EDT | 20,827 |
| Finnish (fi) | TDT | 12,217 |
| French (fr) | GSD | 14,554 |
| German (de) | GSD | 13,814 |
| Hebrew (he) | HTB | 5,241 |
| Hindi (hi) | HDTB | 13,304 |
| Indonesian (id) | GSD | 4,477 |
| Italian (it) | ISDT | 13,121 |
| Korean (ko) | GSD | 27,410 |
| Latin (la) | PROIEL | 15,906 |
| Latvian (lv) | LVTB | 5,424 |
| Norwegian (no) | Bokmaal | 29870 |
| Polish (pl) | LFG | 19,874 |
| Romanian (ro) | RRT | 8,043 |
| Russian (ru) | SynTagRus | 48,814 |
| Slovak (sk) | SNK | 8,483 |
| Spanish (es) | AnCora | 28,492 |

Table 9: The cross-lingual UAS/LAS results on 22 languages of UD Treebanks.

| *Cross-Transfer* | en | bg | ca | cs | nl | et | fi | fr |
|---|---|---|---|---|---|---|---|---|
| m-BERT | 92.52 / 91.29 | 83.80 / 72.77 | 79.60 / 69.00 | 73.95 / 61.23 | 77.76 / 69.02 | 73.11 / 50.90 | 75.71 / 54.96 | 83.18 / 70.76 |
| **UD-BERT** | 93.01 / 91.43 | 89.64 / 79.02 | 87.18 / 76.06 | 86.07 / 70.38 | 86.79 / 78.74 | 82.38 / 58.53 | 83.86 / 60.97 | 89.07 / 75.86 |
| XLM-R$_{large}$ | 93.10 / 91.32 | 88.67 / 77.91 | 85.00 / 73.71 | 77.80 / 65.12 | 82.10 / 72.79 | 78.66 / 57.67 | 80.97 / 60.63 | 88.27 / 74.71 |
| **UD-XLM-R$_{large}$** | 94.19 / 92.54 | 92.66 / 82.15 | 90.76 / 79.35 | 88.98 / 75.67 | 90.94 / 82.33 | 87.57 / 65.90 | 90.09 / 68.95 | 93.19 / 78.93 |

| | en | de | he | hi | id | it | ko | la |
|---|---|---|---|---|---|---|---|---|
| m-BERT | 92.52 / 91.29 | 76.54 / 66.00 | 73.48 / 47.26 | 43.77 / 29.83 | 57.06 / 47.82 | 87.41 / 80.93 | 36.97 / 23.44 | 52.39 / 36.42 |
| **UD-BERT** | 93.01 / 91.43 | 84.54 / 74.88 | 87.33 / 55.98 | 66.99 / 42.82 | 76.18 / 62.37 | 92.74 / 86.59 | 53.89 / 36.94 | 71.66 / 51.91 |
| XLM-R$_{large}$ | 93.10 / 91.32 | 81.65 / 71.34 | 73.86 / 48.66 | 48.08 / 31.66 | 60.57 / 51.32 | 91.02 / 84.31 | 40.64 / 25.50 | 60.95 / 42.44 |
| **UD-XLM-R$_{large}$** | 94.19 / 92.54 | 88.05 / 79.34 | 86.42 / 56.69 | 65.57 / 47.13 | 77.65 / 64.52 | 95.39 / 88.83 | 52.98 / 37.52 | 76.26 / 57.14 |

| | en | lv | no | pl | ro | ru | sk | es |
|---|---|---|---|---|---|---|---|---|
| m-BERT | 92.52 / 91.29 | 76.51 / 54.52 | 87.28 / 78.67 | 88.76 / 76.43 | 75.94 / 62.83 | 72.73 / 62.40 | 79.14 / 67.86 | 80.08 / 70.64 |
| **UD-BERT** | 93.01 / 91.43 | 83.07 / 60.94 | 90.41 / 82.22 | 92.97 / 81.26 | 87.50 / 72.59 | 85.80 / 72.58 | 88.69 / 74.83 | 85.93 / 76.15 |
| XLM-R$_{large}$ | 93.10 / 91.32 | 83.94 / 62.52 | 90.50 / 82.56 | 90.68 / 77.90 | 81.75 / 69.01 | 76.64 / 66.58 | 81.63 / 70.60 | 85.00 / 74.36 |
| **UD-XLM-R$_{large}$** | 94.19 / 92.54 | 91.21 / 69.64 | 94.14 / 86.37 | 95.48 / 83.91 | 91.28 / 77.80 | 91.29 / 80.64 | 92.02 / 80.56 | 90.37 / 79.78 |

Table 10: The monolingual UD parsing result (UAS/LAS) on treebanks with training set under 100 sentences: Kurmanji(kmr)-MG, Swedish_Sign_Language(swl)-SSLC, Livvi(olo)-KKPP, Kazakh(kk)-KTB, Buryat-BDT(bxr), Upper_Sorbian(hsb)-UFAL.

| *All-FT* | kmr-mg | | swl-sslc | | olo-kkpp | | kk-ktb | | bxr-bdt | | hsb-ufal | |
|---|---|---|---|---|---|---|---|---|---|---|---|---|
| | UAS | LAS | UAS | LAS | UAS | LAS | UAS | LAS | UAS | LAS | UAS | LAS |
| *w/o PrLM* | 18.33 | 8.06 | 19.33 | 10.86 | 41.84 | 30.25 | 66.80 | 58.73 | 39.61 | 17.33 | 68.41 | 53.05 |
| m-BERT | 32.26 | 11.09 | 34.61 | 20.15 | 59.93 | 36.44 | 78.26 | 62.80 | 51.56 | 23.06 | 73.16 | 60.92 |
| **UD-BERT** | 38.80 | 17.01 | 39.82 | 23.50 | 64.48 | 41.71 | 81.13 | 65.56 | 55.68 | 26.29 | 75.33 | 62.94 |
| XLM-R$_{base}$ | 74.62 | 59.86 | 50.87 | 36.81 | 70.21 | 51.52 | 80.03 | 65.20 | 55.24 | 25.95 | 80.17 | 69.92 |
| **UD-XLM-R$_{base}$** | 78.55 | 62.92 | 55.88 | 42.00 | 75.97 | 54.33 | 81.74 | 66.79 | 59.37 | 29.40 | 83.25 | 72.88 |
| XLM-R$_{large}$ | 77.89 | 62.24 | 55.67 | 42.02 | 72.89 | 53.01 | 82.15 | 66.97 | 55.30 | 26.27 | 81.38 | 71.17 |
| **UD-XLM-R$_{large}$** | 81.62 | 64.10 | 58.26 | 43.72 | 76.08 | 56.41 | 85.69 | 70.98 | 60.40 | 30.92 | 85.12 | 74.16 |