# OpenReview forum: "Multilingual Pre-training with Universal Dependency Learning"
_NeurIPS.cc/2021/Conference — NeurIPS 2021 Poster_

### Official Review · Reviewer_JRfY · 2021-07-13

**Rating:** 4
**Confidence:** 4

**Summary:**

This paper presents a multilingual joint language model and syntactic
dependency parser. The model consists of a multilingual transformer
model (BERT or XLM) with a deep biaffine parser (Dozat & Manning,
2017). The outputs of each are then combined to form a final
token-level output layer. Although prior work has already explored the
incorporation of additional structution into a language model
(e.g. StructBERT and LIMIT-BERT), this paper differs by applying its
new model to the multilingual transfer setting and by incorporating
the scores of the syntactic parser back into the embedding
representation. Training mirrors prior work by jointly optimizing a LM
objective and a parsing objective -- the latter on supervised
data. Experiments on XNLI, XQuAD, UD dependency and consituency
parsing, show that the new model achieves improvements over the non-structured baseline transformers.

**Limitations And Societal Impact:**

Mostly, but the paper makes no mention of the (possible) limititations of the model with regard to languages that did not appear in the original supervised parsing data.


**Main Review:**

Strengths:
- Overall the paper is clearly written and offers two extensions beyond prior work:
  1. The output scores of the parser are combined with the language model to produce the final token embeddings. This is achieved by parsing at the token level (rather than the word level as in LIMIT-BERT).
  2. Applying the model to a multilingual setting.
- Although these novelty of the technical contributions are not very grand, the results are very compelling. Improvements are achived on XNLI, XQuAD, and UD dependency/constituency parsing.

Weaknesses:
- The primary ommission is any empirical comparison with the most similar prior work: StructBERT and LIMIT-BERT. Like the UD-BERT/XLM model proposed in this paper, these prior efforts to incorporate linguistic information in pre-training may have implicitly learned useful linguistic structure information within the shared parameters (even though the sharing is less explicit than in this paper). As such, there remains an open question: does feeding the structure back in as done in this paper outperform the approaches taken in StructBERT and LIMIT-BERT? Although these models were only applied to English, this model could straightforwardly be applied in the same setting. (Note that while the ablation experiments in this paper suggest that this feeding back approach is effective, it might be that there are other aspects of those models that provide some benefit not realized in the present model.)
- Other differences between UD-BERT/XLM (this paper) and Struct-BERT and LIMIT-BERT are not addressed. For example, is parsing at the token level (as in this paper) advantageous over the word-level approach of LIMIT-BERT? Is the sentence structural objective of Struct-BERT complimentary to the parsing objective in this paper?
- One of the main claims of this paper is that linguistic structure information is encoded by the model. However, no effort is made to analyze how the linguistic information affects the output embeddings. What does the summation approach taken by this paper accomplish? Were any alternatives considered for how to linearize the dependency scores?
- The paper does not address the impact on languages not included in the supervised parsing data. The model trains the parser submodel on 60 UD languages and tests on 22 UD languages. The language model is trained on 104 languages from Wikipedia. How does the model perform on languages that were *not* included in the original 60 UD training languages? For example, does performance improve or degrade on those languages which are truly low-resource?


**Time Spent Reviewing:**

1.75

---

> ### Author Response · Authors · 2021-08-10
> **Response to Reviewer #4**
>
> About Weakness #1
>
> * Thanks for your suggestion. We will definitely add references and comparisons to StructBERT and LIMIT-BERT. Compared to LIMIT-BERT, the main difference is that we do not regard parsing as a pre-training objective and wish the model learn the syntactic information implicitly, but use the parsing component as an intermediate structure to explicitly learn and extract of the syntactic structure, thereby reducing the black box characteristics of the model. As you have mentioned, we demonstrated in the ablation that structural integration is more significant than simply sharing weights. Compared with StructBERT, although it is related to language structure, StructBERT constructs pre-training objectives through shallow word order and sentence order, and we use artificially annotated syntactic structure as training data, so our model can be directly output the syntactic structure, which is a richer linguistic structure information; in addition, we focus on this universal syntactic structure to improve cross-lingual representation.
> \
> \
> The reason why we didn't directly compare StructBERT and LIMIT-BERT to our work is that our purpose is not to improve monolingual pre-training, but to consider universal syntactic structures in multilingual models to enhance cross-lingual universal representation. Despite this, we still agree with your opinion. Using additional pre-training objective like in StructBERT and LIMIT-BERT on the model may also further improve the model, but because pre-training is very time-consuming and resource-consuming, we will add this ablation experiment in the future version.
>
> About Weakness #2
>
> * Thanks for your question. Although very interesting, it is far from the research theme --- the cross-lingual universal syntax-aware representation. We will supplement additional discussions in the related work.  If only compared in monolingual PrLM aspect, LIMIT-BERT converted WordPiece sequence vector to word-level by simply taking the last token vector of the word as the representation of the whole word. As a result, many token representations are not directly improved during the pre-training of structure knowledge. In addition, since the structure integration layer we proposed directly integrates the structure knowledge into the representation, integrating the same information into all tokens of a word is obviously not good enough (eg: not applicable to the training process of token level MLM). Parsing at the token level enables us to learn and integrate much richer syntax knowledge. Regarding whether StructBERT's sentence structural objective can help our model, we think the answer is yes. Because our method is within sentences, it does not conflict with the objectives across sentences.
>
> About Weakness #3
>
> * Thanks for your valuable questions. In fact, in our ablation experiments, we have already studied the effects of syntactic features integration. We do adopt the summation integration approach. Since this approach does not affect the dimensions and will not bring additional changes to the subsequent structure, we adopt this approach instead of other methods such as concatenation. We can study this kind of details, but because our work is on a large pre-trained language model, it is very time-consuming and resource-consuming, and this choice requires a lot of time. Therefore, we usually tend to learn from common practices in some other work instead of directly enumerating all of them, and other experiments show that the difference between different feature integration methods is not big. In response to your question, our work is not to line-sequence the dependency trees, but to integrate features. Of course, the product of the dependency syntax tree can also be added to the encoder for encoding, but we believe that this complicated method is not as simple and easy to use as ours.
>
> About Weakness #4
>
> * Thanks for tour question. We believe that a large number of annotations of different languages in UD can inspire low-resource languages that are not included. This assertion can be validated with UD parsing, because some of the languages (or treebanks) in UD do not contain annotated training sets, so we did not add their test sets into pre-training. We will individually list the evaluation results on some language without annotated UD training set in the Appendix.
> \
> By the way, our primary aim is not to solve the problem of low-resource languages, but to obtain a universal representation through universal dependency syntax in order to improve the cross-lingual transfer ability in multilingual PrLMs.

---

### Official Review · Reviewer_kc6w · 2021-07-14

**Rating:** 5
**Confidence:** 4

**Summary:**


This paper proposes to include syntactic supervision from the Universal Dependency (UD) dataset for pre-training Transformer-based multilingual encoders, in addition to the typical masked language modeling objective.

The proposed method not only utilizes the parsing objective to guide the training of a portion of the model parameters but also incorporates the features computed for parsing in the process of deriving the final contextualized word representations.

The resulting multilingual pre-trained language models are evaluated on cross-lingual NLI & QA in addition to mono/cross-lingual constituency/dependency parsing.
The paper claims that the newly introduced pre-training strategy is superior to that of the existing baselines in most cases.

**Limitations And Societal Impact:**


Please refer to the Weaknesses part in the main review.


**Main Review:**



### Strengths
- According to the paper, the proposed method shows its effectiveness in diverse downstream tasks such as cross-lingual NLI & QA and mono/cross-lingual constituency/dependency parsing.
- The proposed approach attempts to directly leverage the features derived from the parsing objective to generate the final representations that can consider syntactic aspects of input sentences.


### Weaknesses
- A more clear and accurate method specification is needed (Section 3.3): It is hard to understand the provided equations including one containing softmax.
For instance, in the softmax function, the first term $R^T_{dep}U_1R_{head}$ should be a matrix while the second and third terms should be vectors. I don't understand how you can directly add that matrix and those vectors without extra processing.
Furthermore, it is unclear in the paper how the terms $R_m$ and $S^k$ and the final universal structure learning loss ($L_{USL}$) can be connected.
Please correct this if I'm wrong.
It seems that the formulations are too simplified to clearly state their meaning.
- An existence of confounding factors that can have an impact on performance: To the best of my knowledge, the proposed method appends additional parsing-specific and final integration modules on top of the typical Transformer architecture which is widely used for pre-training. This implies that the models proposed by this work naturally have more parameters than existing baselines such as m-BERT and XLM-R (therefore, please report the number of parameters of each approach).
Furthermore, the proposed models can see more **plain** sentences from the UD dataset in addition to getting access to structural supervision from parse trees.
Considering the aforementioned confounding factors, I'm not sure the increase in performance is entirely due to the effectiveness of structural supervision.
Although the ablation study tried to mitigate some related concerns, the claim of the paper would be much more persuasive if additional experiments can be conducted in a more rigorous condition, e.g., adjusting the number of parameters of each model to a comparable extent.


### Questions to the authors
- In the dependency parsing experiment, the authors decided to train a separate parser instead of using one obtained during pre-training. I'm wondering it's because the pre-trained model cannot be utilized as an effective parser or there exist another reasons in the experiment setup. If we should train a separate parser for performance even in the case where we already included our training dataset in the pre-training phase, I'm not sure the proposed method is attractive especially in terms of data efficiency.
- Can you imagine some tasks (other than ones presented in the paper) that can be much more benefitted from explicitly considering syntactic information? The experimental results reported in the paper seem to be a little bit incremental, and therefore, if you can find such a proper application, I believe that the contribution of this paper would be much more valuable.



**Time Spent Reviewing:**

4

---

> ### Author Response · Authors · 2021-08-10
> **Response to Reviewer #3**
>
> About Weakness #1
>
> * Thanks for your suggestion. Actually, there are some extra processing, but since this part is not our innovation, we did not write it in detail. Please allow me to explain it in detail here:
>
>     Taking $S^{label}$ as an example, $R^T_{dep}U_1R_{head} \in \mathcal{R}^{|D|\times H_{dep} \times H_{head}}$, $u_2^TR_{head} \in \mathcal{R}^{|D|\times H_{head}}$, $u_3^TR_{dep} \in \mathcal{R}^{|D|\times H_{dep}}$ and we first need to unsqueeze the dimensions of the latter two terms to $\mathcal{R}^{|D|\times 1 \times H_{head}}$ and $\mathcal{R}^{|D| \times H_{dep} \times 1}$ and then add them up together with the bias $b$.
> * For your question about $R_m$, $S^k$ and $\mathcal{L}^{USL}$: $R_m$ ($R_{head}$ and $R_{dep}$) represent the head and dependency representation obtained from feeding the output of encoder into two separate MLPs. For calculating $S^{arc}$ and $S^{label}$, we both need a pair of $R_{head}$ and $R_{dep}$. $S^{arc} \in \mathcal{R}^{H_{dep}\times H_{head}}$ is the arc score of each token in a sentence, in detail, $S_{ij}^{arc}$ represents the probability that the token $i$ is the head of the token $j$ (equal to $P(y_j^{arc}=x_i|x_j)$). $S^{label} \in \mathcal{R}^{|D|\times H_{dep}\times H_{head}}$ is the label score of each label and each token in a sentence, in detail, $S_{kij}^{label}$ represents the probability that the token $i$ is the head of the token $j$ with label $k$ (equal to $P(y_j^{label}=k|x_j,y_j^{arc}=x_i)$. Once we have these probabilities, we can use that formula for $\mathcal{L}^{USL}$ to calculate the loss.
>
> About Weakness #2
>
> * Thanks for your suggestion. The comparison of number of parameters between our full model and the baselines as well as the data statistics of our training data are as follows:
>          UD-BERT: 192.0M    m-BERT: 177.9M
>          UD-XLM-R base: 291.8M    XLM-R base: 278.0M
>          UD-XLM-R large: 581.06M    XLM-R large: 559.9M
>          UD: 1.06M (sents)    Wiki: 372.1M (sents)    CommonCrawl: 2501.2M (sents)
> In our current version, when our model only uses the model structure without UD pre-training (*w/o UD Pre-training*), the performance is decreased compared to the full model but the number of parameters and the training data for MLM keep the same. Admittedly, the number of parameters can affect the model, but the structural knowledge we integrate is also very effective. And we believe that appropriate parameter increase is necessary due to the additional structure. And according to the results of the ablation, which indicated that the structure integration we proposed was reasonable, and it could better learn the syntactic knowledge without supervision compared with the baseline.
> * Second, about additional UD plain text exposed to the full model, the number of UD plain texts is relatively small compared to the original training data of MLM, so it is not expected to be very useful for tasks other than UD parsing. In addition, we acknowledge that part of improvement of UD parsing did benefit from adding its data to pre-training, but it was fair since we also train the baseline with same update steps.
> * In the revised version, we will select more languages and conduct ablation experiments on different tasks such as XNLI and XQuAD to make it more convincing. We will also add UD plain text into the MLM training set to train the baselines for comparison, so as to more clearly show the improvement brought by structural supervision, UD plain text, structural integration and other aspects to the model.
>
> About Question #1
>
> * Thanks for your valuable question. Actually, using our model to directly parse on the test sets of UD can get very good results which are comparable to the current SOTA results. Because the parser inside our language model is subword-level while the compared related works are all word-level, to make the comparison more fair and training a separate parser could improve the results even further, so we showed this results in our experimental section. We will report the results of syntactic evaluation directly on the test set of each language after the model pre-training instead of further parsing training in the Appendix.
>
> About Question #2
>
> * Thanks very much for your question. Our aim is to create a multilingual model that is both a contextualized language model and a syntactic parser.
> On the one hand, the syntax-aware language representation obtained in this way can be directly applied to end-to-end downstream tasks without the need for additional syntactic encoders, so any downstream tasks can easily integrate syntactic information and a strong pre-trained language model at the same time.
> On the other hand, our model can not only provide representation features to enhance the downstream neural model, but also provide syntactic parses for rule-based models rather than depends on multiple upstream models, for example, knowledge graphs construction.

---

### Official Review · Reviewer_ZFQx · 2021-07-15

**Rating:** 7
**Confidence:** 3

**Summary:**

This paper proposes to pre-train multilingual language models with the knowledge of dependency parsing. The universal structure knowledge is used in two ways, the learning objective and improvement for representation learning for mask language model. The effectiveness of proposed approach is verified for two pre-trained models, mBERT and XLM on  several downstream tasks.

**Limitations And Societal Impact:**

Yes

**Main Review:**

Although the idea of leveraging syntactic structure knowledge is not new, this paper proposes a reasonable and interesting way to achieve that goal – multi-task learning with explicit learning objective and improvement for representation learning by leveraging the hidden states of dependency parsing.

The authors verify its effectiveness on XNLI and XQuAD based on BERT and XLM-R, and it also shows improvement on the linguistic structure parsing task.

Overall speaking, the UD-aware approach for multilingual pre-training looks interesting and experimental results are convincing.



**Time Spent Reviewing:**

3

---

> ### Author Response · Authors · 2021-08-10
> **Response to Reviewer #2**
>
> * Thank you very much for the comments and recognition. Our contribution is not only in the improvement of the performance of the language model by the monolingual syntax information, but also in the enhancement of the universal syntax for cross-lingual universal representation.

---

### Official Review · Reviewer_uazG · 2021-07-22

**Rating:** 6
**Confidence:** 4

**Summary:**

The paper proposes to pre-train multilingual encoders explicitly on dependency parsing and language modeling to improve cross-lingual transfer learning. The proposed method, UD-BERT composed of the Transformer encoder along with a universal structure learning layer and a universal structure integration layer. The experiment results show the promise of the proposed method.

**Limitations And Societal Impact:**

I do not see any discussion, perhaps it is not required.

**Main Review:**

**Originality**

The proposed idea is not new, however, it probably is not investigated for multilingual representation learning.

**Quality and Clarity**

The work has a balance between modeling and experiments. The writing is clear; all the model components, experiment settings, experiment tasks, and results are sufficiently explained.

**Significance**

This paper investigates the advantage of using the universal linguistic structure to improve cross-lingual transfer. The proposed technique has a value, the empirical results show improvements over the baselines.

**Weaknesses**

I see one weakness in the experiments which forced me to give a lower score to this paper. The evaluation results show a very marginal improvement over the baseline. I am worried this could be due to variances across experiments. It seems like the authors didn't perform the experiments with different random seeds. For multilingual experiments, it is highly recommended to run experiments multiple times and report the average scores.



**Time Spent Reviewing:**

4

---

> ### Author Response · Authors · 2021-08-10
> **Response to Reviewer #1**
>
> About Originality
>
> * Thanks for your comments. Existing monolingual PrLMs almost take the language structure knowledge as a feature or pre-training objective. Our work not only extends this exploration to multilingual representation learning, but also innovatively proposes an approach of syntactic structural integration and universal representation based on the universal syntax, which has been shown to help models better learn and utilize syntactic knowledge in both monolingual and cross-lingual situations in our experiments. It is worth noting that the focus of our method is on universal representation, rather than integrating syntactic information only.
>
> About Weakness
>
> * Thanks for your comments. Actually, our reported evaluation results are already the average of 5 different random seeds on XNLI and XQuAD. Besides, these average improvements were greater than 0.5%, which is with a statistically significance. We will explain this explicitly, and mark the range of results in the Appendix due to the space constraints.

---

### Author Response · Authors · 2021-08-10
**General Response**

We thank the reviewer for their thorough reviews and insightful feedback! We are especially grateful to comprehensive suggestion on improving the presentation of related work. Below we provide some clarifications to answer questions of general interest in detail. We will improve our work in response to these questions and other suggestions you have mentioned.

**Systematically summarize the innovation of our work**

* First, compared with existing monolingual PrLMs most of which take the language structure knowledge as a feature or pre-training objective, we propose an innovative approach of syntactic structural integration and universal representation based on the universal syntax, which has been shown to help models better learn and utilize syntactic knowledge in both monolingual and cross-lingual situations in our experiments. This approach allows us to focus on universal representation rather than integrating syntax information. Specifically, compared with the previous work of monolingual PrLM+Syntax, we regard parsing not only as a pre-training objective, but also as an active part of the overall PrLM structure, so this part of the structure will further exert its effectiveness in downstream tasks rather than discarding it in the finetune stage, which increases the adaptability from pre-training to finetune phases.

* Second, the multilingual PrLMs suffer a lot from lacking explicit alignment signals across languages, although joint training is performed in the same model, the unsupervised method makes cross-lingual transfer inefficiency and keeps the learning still challenging. Some works explicitly use parallel bitext as a high-level alignment signal. However, the amount of parallel bitext is small and are difficult to obtain. Our work instead focuses on the universal dependency structure which is a relatively low-level linguistic alignment knowledge. The raw text of each language thus can exhibit its own unique linguistic traits, while the consistent syntax across languages can serve as the anchor points across multiple languages, which can greatly facilitate cross-lingual representation transfer.
For languages which do not have available UD annotations, because the syntactic similarities of different languages are obvious different and our model has integrated syntactic structure knowledge of multiple languages, our model can also find a relatively accurate hypothetical anchor point and give a good representation. Due to time constraints, we will report the evaluation results of our model on languages which have no annotated UD training sets after rebuttal.

**Supplements to the experiment**

* Allowing for a fair comparison to the baseline, rather than taking additional training steps based on the original weights of the baseline, we pre-trained our model from scratch which takes a lot of time and resources to implement massive experiment on our quite limited computational resources. As a result, while our ablation study has a lot to explore, we only chose two languages and a single evaluation task. In the rebuttal stage, by meeting the requirement and suggestions from reviewers, we have done more ablation experiments on XQuAD for 4 languages $en$, $ar$, $de$ and $el$, and the results are listed in the table below (the evaluation criterion is F1/EM). The results show that UD knowledge can indeed help the model improve the cross-lingual transfer ability of universal representation, and our structure integration plays a decisive role in making the model learn and integrate universal structure knowledge (language alignment signals) more effectively, both supervised and unsupervised. After rebuttal stage, we will further reinforce our ablation experiment and update it in the later version.
|        |  m-BERT   | UD-BERT  | w/o Universal Structure Integration | w/o UD Pretraining  |
|  :----:  |  :----:  | :----:  | :----:  | :----:  |
| $en$  | 83.5/72.2 | 83.9/72.5 | 83.8/72.6 | 83.6/72.2 |
| $ar$  | 61.5/45.1 |62.8/46.3 | 61.9/45.4 | 61.5/45.0 |
| $de$  | 70.6/54.0 |72.3/56.0 | 70.9/54.5 | 70.7/54.3 |
| $el$  | 62.6/44.9 |62.9/45.8 | 62.5/45.0 | 62.7/45.2 |

* Although we didn't mention it in the paper, our reported evaluation results are already the average of 5 different random seeds on XNLI and XQuAD. Besides, these average improvements were greater than 0.5%, proving that our improvements were statistically significant.
Because listing the range of results requires a large table, we will explain this explicitly, and mark the range of results in the Appendix after rebuttal.
\
In the rebuttal stage, we sampled 100 sentences in the test set of each languages as the final parsing test set (less than 100 take all), a total of 9767 sentences, and directly evaluated the parser performance of our $\text{UD-XLM-R}_{\text{large}}$ model (without finetune), and the result is as follows:
         UAS: 91.1 LAS: 86.4
We can see that the UAS of our model reaches 91.1, which is not much different from the average score we get from retraining a parser in 22 languages, indicating that our model can be treated as a practical parser. In addition, the final parsing test set includes languages and treebanks that do not have annotated training sets, which confirms that our model is also helpful for low resource languages.

* Since it is pressed for time and the PDF cannot be updated, we cannot report more experimental results here. However, we will improve our work in the coming time according to your valuable comments and add new results in our future version.

---

### Author Response · Authors · 2021-08-26
**General Response #2**

We thank all the reviewers for the insightful feedback with the score increased. Below we make some responses to the comments we received recently.
While we have clarified the innovation of our work in our previous response, if we may, let us emphasize it again here by comparing our work to recent related researches mentioned in the latest review feedback.

LISA [1] is a Transformer model that uses syntax information to enhance SRL. It incorporates syntactic information by training one attention head predicting syntactic dependency arc. First of all, LISA only aims at one specific task, SRL, which is far from our research goal towards the general-purpose pre-training which supposes to benefit all possible NLP tasks, so it is doubtful whether its method can be applied to the PrLM and improve the performance of multiple tasks and arc prediction with only one attention head is difficult to achieve high enough accuracy. Secondly, LISA can only use the arc feature of syntax but not the relations, while our approach of syntactic structural integration can integrate both of the arc and relation features learned by our model into the final universal representation. By the way, our PrLM can be used directly as a well-behaved parser, but LISA clearly does not.

Sachan et al. [2] investigate two distinct syntax-infused Transformers which obtain state-of-the-art results on SRL and relation extraction tasks. However, it also reveals a critical shortcoming of these models: their performance gains are highly contingent on the availability of human-annotated dependency parses, which raises important questions regarding the viability of syntax-augmented Transformers in real-world applications. Fortunately, we have just the right solution to this problem. Our ablation shows that our universal structure integration can help the model learn the syntactic knowledge without additional supervision and improve the performance because it has an accountable form that can well model the syntactic structure features.

Recently, Ahmad et al. [3] propose a multilingual PrLM training mBERT using an auxiliary objective to encode the universal dependency tree structure that helps cross-lingual transfer.
However, our work significantly differs from theirs. We regard parsing not only as a pre-training objective (as nearly all existing work did including [3]), but also as an active part of the overall PrLM structure, so this part of the structure will further exert its effectiveness in downstream tasks rather than discarding it in the finetune stage, which increases the adaptability from pre-training to finetune phases. Actually, our UD-BERT outperforms [3] on the same tasks such as XNLI and XQuAD, which exactly shows our multilingual syntax+pre-training solution is a better choice than the previous work.

[1] Linguistically-Informed Self-Attention for Semantic Role Labeling. Emma Strubell, Patrick Verga, Daniel Andor, David Weiss, Andrew McCallum. EMNLP, 2018.

[2] Do Syntax Trees Help Pre-trained Transformers Extract Information? Devendra Sachan, Yuhao Zhang, Peng Qi, William L. Hamilton. EACL, 2021.

[3] Syntax-augmented Multilingual BERT for Cross-lingual Transfer. Wasi Uddin Ahmad, Haoran Li, Kai-Wei Chang, Yashar Mehdad. ACL, 2021.

---

### Decision · Program_Chairs · 2021-09-28

**Decision:**

Accept (Poster)

**Comment:**

This paper presents an approach to multilingual pretraining that (a) incorporates supervised dependency parsing as an auxiliary objective and (b) incorporates dependency scores back into the encoder itself. Results on XNLI, XQuAD, and UD dependency and constituency parsing show gains over baselines that do not use syntactic structure. Reviewers are split with two in favor of rejection and two voting for acceptance. Most reviewers viewed both the proposed approach and the positive results as potentially impactful. However, a serious concern was surfaced. Specifically, that the experimental comparisons conflate the two potential contributions: (1) gains from incorporating parsing scores back into the encoder, and (2) gains from training in a multilingual setting. These points need to be separately evaluated in the context of missing, but closely related baselines. For example, how does the method compare with related baselines like Struct-BERT or LIMIT-BERT that also consider syntactic structure, but in a monolingual setting? Does incorporating parsing scores back into these encoders increase performance in a monolingual setting? Do these baselines improve in a multi-lingual setting without other architectural changes? These concerns sway me to recommend rejection for the current draft. But I strongly encourage authors to resubmit with additional experiments.

**Consistency Experiment:**

NeurIPS has a long history of experimentation. In 2014, NeurIPS ran an experiment in which 10% of submissions were reviewed by two independent committees to quantify the randomness in the review process. This year, we repeated a variant of this experiment to see how the quality of the review process has changed over time.  This paper was part of the experiment and was therefore assigned to two committees (consisting of reviewers, an Area Chair, and a Senior Area Chair) that reached independent decisions.  If both committees made the same recommendation, this recommendation was followed. If a single committee recommended acceptance, the paper was accepted (with the exception of a few cases in which the other committee identified what we considered a fatal flaw, e.g., an error in a key result).

This copy’s committee reached the following decision: **Reject**

The other committee assigned to the paper recommended **Accept (Poster)**.  You can find the other set of reviews, along with any follow up discussion with the authors here:
https://openreview.net/forum?id=5JIAKpVrmZK